# Influence of Certain Natural Bioactive Compounds on Glycemic Control: A Narrative Review

**DOI:** 10.3390/nu18010052

**Published:** 2025-12-23

**Authors:** Marta Pelczyńska, Starosta Szymon, Michał Konieczny, Hubert Bączyk, Jakub Szyszko, Krzysztof Cholewa, Paweł Bogdański

**Affiliations:** 1Department of Treatment of Obesity, Metabolic Disorders and Clinical Dietetics, Poznan University of Medical Sciences, 49 Przybyszewskiego Street, 60-355 Poznan, Poland; mpelczynska@ump.eu.pl; 2Faculty of Medicine, Poznan University of Medical Sciences, 70 Bukowska Street, 60-812 Poznan, Poland; 91423@student.ump.edu.pl (S.S.); 91285@student.ump.edu.pl (M.K.); 91169@student.ump.edu.pl (H.B.); 91448@student.ump.edu.pl (J.S.); 91836@student.ump.edu.pl (K.C.)

**Keywords:** glycemic control, bioactive compounds, mannoheptulose, β-carotene, resveratrol, steviosides, curcumin

## Abstract

Glycemic control disorders, including insulin resistance (IR) and type 2 diabetes (T2D), represent a major global health challenge. Although existing therapeutic strategies demonstrate effectiveness regarding glycemic control and reduction in diabetes-associated mortality, they are often associated with limited patient tolerance and adherence. Consequently, there is growing interest in natural bioactive compounds that may support glycemic regulation while potentially posing a lower risk of adverse effects in ongoing therapy. The objective of this review is to evaluate the potential of selected natural substances in the context of blood glucose regulation. The analysis encompasses data from in vitro, in vivo, and clinical studies on compounds such as mannoheptulose, β-carotene, resveratrol, steviol glycosides, and curcumin. These agents have demonstrated the ability to modulate key metabolic pathways, enhance tissue insulin sensitivity, reduce oxidative stress, and support pancreatic β-cell function. Particularly promising effects have been observed when some of these compounds are combined with conventional antidiabetic medications, such as metformin. The review also highlights relevant molecular mechanisms, including activation of the AMP-activated protein kinase (AMPK) pathway, increased expression of glucose transporter type 4 (GLUT4), and modulation of gene expression related to insulin sensitivity. Despite encouraging findings, further clinical research is necessary to determine optimal dosages, therapeutic protocols, and the long-term safety of these substances in human populations. Natural bioactive compounds may thus represent a valuable adjunct to current strategies for managing glycemic disorders.

## 1. Introduction

Disorders of glycemic control represent a significant health issue affecting both older and younger populations. As the prevalence of glycemic disorders increases, there is also a rising number of deaths associated with T2D, partly due to the limitations of current therapeutic strategies regarding patient adherence and tolerance, with epidemiological projections suggesting that the number of adult individuals with carbohydrate disorders would reach as many as 700 million by 2045. T2D is characterized by impaired pancreatic β-cell function, leading to insufficient insulin secretion and elevated peripheral blood glucose levels. It is often exacerbated by coexisting insulin resistance (IR) in target tissues. The etiology of T2D is multifactorial, involving both genetic predispositions and environmental factors, such as a Western diet, excessive body weight, and low physical activity [1]. A crucial aspect of modern dietary habits is the growing consumption of highly processed foods, which are rich in simple sugars and saturated fatty acids, leading to excessive calorie intake and an increased prevalence of obesity [2]. Currently, caloric restriction (CR) is considered an effective strategy for preventing and treating excessive body weight, often correlated with carbohydrate dysregulation. However, adherence to strict dietary guidelines often proves challenging for patients with established poor dietary habits. This has led to the concept of caloric restriction mimetics (CRMs), substances that replicate the metabolic, hormonal, and physiological effects associated with reduced calorie intake without requiring significant decreases in food consumption. These mimetics activate adaptive mechanisms in the body, enhancing resistance to oxidative stress, reducing the risk of age-related diseases, and helping maintain physiological functions. Mimetics, which can occur naturally or be synthetic, represent a broad range of bioactive compounds, including substances such as mannoheptulose (MH) and resveratrol. The substances included in this review, that is β-carotene, stevia glycosides, and curcumin, are considered natural compounds with antihyperglycemic effects [3]. In this review, CRMs are not presented as a substitute for lifestyle modification, which remains the cornerstone of preventing and managing glycemic disorders. Rather, they are considered a complementary approach in circumstances where full adherence to dietary and physical activity recommendations is difficult to achieve. It can be concluded that caloric restriction and health-promoting dietary patterns elicit favorable metabolic adaptations, including improved insulin sensitivity, reduced oxidative stress, and enhanced mitochondrial function. CRMs aim to reproduce selected beneficial effects of caloric restriction without necessitating substantial reductions in energy intake [3].

Bioactive compounds are simple or complex nutrients that can interact with targeted tissues by presenting a wide range of beneficial effects to the human body. They are usually obtained from plants such as fruits, vegetables, and spices (especially from a group of polyphenols). There are different mechanisms associated with the prevention of chronic disease, especially civilization-linked disorders, through the consumption of bioactive compounds from the diet. They include an antioxidative capacity, anti-inflammatory and antiproliferative properties, as well as the modulation of the expression of certain genes [4]. Thus, it seems that bioactive compounds may also play an important role in the regulation of glucose homeostasis [5]. These agents have demonstrated the ability to modulate carbohydrate metabolism by reducing fasting and post-meal hyperglycemia, enhancing tissue insulin sensitivity, and supporting pancreatic β-cell function. Overall, they represent a novel approach to support the prevention and treatment of glucose dysregulation [6,7].

The objective of this narrative review is to identify and critically assess certain naturally occurring bioactive compounds with potential for regulating blood glucose levels. Analysis of available in vitro, in vivo, and clinical studies aims to highlight substances with the greatest hypoglycemic potential. Furthermore, this work identifies existing gaps in research and proposes recommendations for future scientific investigations, supporting the development of safe, natural therapies conducive to maintaining proper glycemic control.

## 2. Methodology

The comprehensive overview was conducted between March and July 2025 using the National Library of Medicine browser (Medline, PubMed, Web of Science). The following keywords and medical subject headings (MeSH) were used: glycemia, blood glucose, glycemic control, diabetes, mannoheptulose (MH), β-carotene, resveratrol, steviosides, and curcumin.

The selection of articles was performed according to the following criteria: (i) articles written in English published mostly in the last 25 years (except for a few historical publications with significant medical meaning); (ii) studies focusing on in vitro procedures, animal trails, and adult human subjects study; (iii) articles whose outcomes indicated glucose homeostasis (e.g., fasting blood glucose level, postprandial blood glucose concentration, glycosylated hemoglobin, IR). All the references were manually selected by the authors.

The structure of each section is as follows. First, the general characterization of selected bioactive compounds has been described. Secondly, the in vitro, animal, and human studies were reviewed. In the next sections, the comparison with antidiabetic medications as well as the mechanisms of action between bioactive compounds and glucose metabolism were given. Finally, the conclusion with a summary of the current state of research was formulated.

## 3. Unripe Avocado Extract—Mannoheptulose

### 3.1. General Characterization

Extract from unripe avocado constitutes a rich source of the rare compound—MH, a 7-carbon simple sugar classified as a ketoheptose [8]. Although the avocado extract (AvX) is the most abundant source of MH, it is also present in other plants such as alfalfa, fig tree, and primrose. It is important to note that the MH content decreases as the avocado fruit ripens, which is a significant factor when planning harvests [9]. MH was proposed within the framework of the CRM concept [3]. In contrast to 2-deoxyglucose (a compound that inhibits the activity of phosphoglucose isomerase, the second step of intracellular glycolysis) [10], which is also presented as a candidate for CRM, MH exhibits lower toxicity [11,12,13]. Although MH seems to modulate glucose metabolism, the number of studies on MH is limited, with most being based on animal models [12,14,15]. It is worth mentioning that MH potential as a treatment for hypoglycemia was also considered due to its ability to increase blood glucose levels while significantly lowering insulin levels, as observed in animals receiving a high intravenous dose of MH per kg of body weight [14]. What is important, it has been observed that the capacity for MH absorption varies considerably not only between species (dogs, rats, monkeys, cats, and humans) but also among individual subjects [16].

### 3.2. The Effect on Glucose and Insulin Levels

#### 3.2.1. In Vitro Studies

Ingram et al. study utilized L6 rat muscle cells to investigate the effects of AvX on the expression of proteins associated with CRM. During the experimental procedures, cells were treated with various concentrations of AvX. Analysis of the results demonstrated that exposure of L6 cells to AvX significantly increased the expression levels of proteins involved in the calorie restriction response. A notable increase in the expression of sirtuin 1 (SIRT1), adenosine monophosphate-activated protein kinase-α1 (AMPKα1), and peroxisome proliferator-activated receptor gamma coactivator 1-alpha (PGC-1α) was observed. These patterns correspond to classical signaling pathways activated during calorie restriction, suggesting that AvX may be a promising natural candidate for a calorie restriction mimetic [9].

It should be emphasized that the available in vitro findings do not allow the observed effects to be attributed exclusively to MH. The applied experimental models do not enable a clear distinction between the actions of isolated MH and those of the broader matrix of bioactive constituents present in AvX. This issue becomes particularly relevant in the context of studies demonstrating that components unrelated to MH, including polyphenols, fatty acids, and other secondary metabolites, may also exert significant modulatory effects on carbohydrate metabolism. The study by Younis et al. showed that the Gwen variety of avocado, characterized by a higher content of polyunsaturated fatty acids and polyphenols, exhibited the strongest inhibition of α-glucosidase and α-amylase, with IC_50_ values of 55.07 ± 2.48 μg/mL and 95.3 ± 2.02 μg/mL, respectively, surpassing the efficacy of acarbose, and representing the highest antidiabetic potential [17]. Similarly, Rao US et al. reported that the hydroethanolic extract exhibited the lowest IC_50_ value for α-amylase (0.42 μg/mL), whereas the ethanolic extract showed the strongest inhibition of α-glucosidase (IC_50_ = 0.15 μg/mL). This pattern may suggest that phenolic-rich AvX contributes to the inhibition of carbohydrate-digesting enzymes, representing a promising antidiabetic agent [18]. Furthermore, Ehikioya et al. demonstrated that extracts derived from the leaves, roots, and bark of *Persea americana* exerted strong inhibitory effects on α-amylase and α-glucosidase, with hypoglycemic activity comparable to that of glibenclamide in animal models [19]. Taken together, these observations suggest that the glucose-lowering effects associated with avocado consumption likely result from the synergistic actions of multiple classes of bioactive compounds rather than from a single constituent such as MH. This highlights the need for further studies with purified fractions and standardized extracts to precisely identify which components of avocado are related to observed metabolic effects.

#### 3.2.2. Animal Studies

To date, scientific studies emphasize that the use of MH in combination with another substance (such as gelatin) is essential, as observations indicate that oral administration of pure MH can lead to diarrhea [16]. Despite the diversity of results regarding the influence of MH on glucose metabolism and insulin action, the literature reveals a correlation between these parameters. A study conducted on animals (such as cats and dogs) showed that administering MH at a dose of 8 mg/kg (until 28 days) did not involve significant changes in glucose or insulin levels, although an increase in energy expenditure was observed [20,21]. In subsequent studies, where the duration of MH administration was extended to 50 days at similar doses, a significant decrease in plasma insulin concentration was noted, while glucose levels remained relatively stable [9]. Further analysis of studies using much higher doses of MH (above 1000 mg/kg) in rats revealed that such exposure led to hyperglycemia and a decrease in insulin levels. In the research by Paulsen et al., it was found that administration of MH at a dose of 1000 mg/kg in rats resulted in a rapid and pronounced increase in blood glucose levels [14]. Similarly, according to the study by Viktora et al., it was demonstrated that oral administration of MH in doses ranging from 500 to 8000 mg/kg of body weight led to a dose-dependent increase in glycemia. The highest glucose concentrations were observed approximately two hours after administration, and the severity of hyperglycemia increased with higher doses of MH [16]. Conversely, as reported in the study by Ingram et al., it was shown that mice fed a diet AvX for 12 weeks, delivering approximately 1700 mg/kg of MH per day, exhibited a significant reduction in blood insulin levels and a moderate decrease in glucose levels at the end of the study period [9]. Based on the presented results, it can be concluded that a single high dose of MH causes transient hyperglycemia and a decrease in insulin levels, while its chronic administration leads to a reduction in both blood glucose and insulin concentrations.

#### 3.2.3. Human Studies

The first study on MH conducted in humans (apart from its use as a treatment for hypoglycemia) involved healthy volunteers (n = 8) with no family history of diabetes. During the experiment, the participants consumed slices of fresh avocado fruit, and calculations indicated that the ingested dose of MH ranged from 2.15 to 12.83 g per person (from 33 to 200 mg/kg of body weight). Plasma glucose concentrations remained essentially unchanged, whereas five of the eight subjects showed a marked decrease in immunoreactive insulin levels. Statistical analysis confirmed that the average reduction in insulin levels was statistically significant [16]. It is important to note, however, that unlike other studies, this investigation involved the consumption of fresh avocado fruit slices, thereby raising the question of whether the effects observed from fresh avocado fruit are comparable to those reported with AvX in a previous analysis.

The study by Johnson et al. was conducted in two distinct phases: an initial pilot study and a subsequent therapeutic trial in a patient. In the pilot phase, six healthy volunteers, after an overnight fast, were given an isotonic solution containing MH in three different doses (5, 10, or 20 g administered as a single dose per day). The results indicated that MH administration was associated with a moderate increase in glucose concentration, approximately 15%, observed between 1.5 and 4 h post administration, with this effect diminishing after about 6 h. Importantly, the rise in glucose levels was not accompanied by a simultaneous increase in insulin secretion. It should also be emphasized that the highest dose (20 g) was associated with adverse effects such as nausea and diarrhea, suggesting limitations to its use. In the subsequent part of the experiment, healthy volunteers (n = 12) received a single dose of 10 g of MH or a placebo. Two hours after administration, an intravenous glucose tolerance test, using 0.5 g of glucose per kg of body weight, was performed. It was observed that the initial insulin levels were lower, and the decline in glucose concentration occurred more slowly in the study group. However, these differences did not always reach full statistical significance, suggesting variable metabolic responses among participants [15].

The next study involved adult patients with obesity (n = 60), consuming 10 g of AvX, which provided approximately 190 mg of MH per day (about 2 mg/kg of body weight). Both before and after the intervention, an oral glucose tolerance test (OGTT) was performed. The primary outcome, defined as the change in glucose AUC, did not show significant differences between the AvX and placebo groups. Nevertheless, there was a tendency toward a reduction in insulin AUC in participants supplemented with the extract, which was particularly evident in the subgroup that initially exhibited higher postprandial insulin levels (Table 1) [12].

In recent years, the effects of avocado consumption on carbohydrate metabolism and metabolic parameters have been an area of scientific investigation. Findings from contemporary interventional and observational studies indicate that regular avocado intake may exert beneficial effects on metabolic profiles; however, the outcomes related to glycemic parameters are not always consistent. Khan et al. demonstrated that daily consumption of fresh avocado fruit led to alterations in fat distribution among women with overweight and obesity, although it did not significantly affect insulin sensitivity [22]. In contrast, an analysis conducted by Wood et al. revealed that, within a population of adults of Hispanic/Latino ancestry, fresh avocado fruit consumption was associated with a reduced risk of developing T2D, particularly among individuals with prediabetes [23]. Those results were confirmed by another study [24]. What is more, the latest meta-analysis showed that although fresh avocado fruit intake had no significant effect on fasting blood glucose (WMD: −0.05 mg/dL; *p* = 0.78) and other metabolic parameters (such as BMI, total cholesterol, HDL-cholesterol), it significantly reduced LDL-cholesterol and systolic blood pressure, which indicates its cardioprotective effects [25]. Interesting results were shown in other umbrella reviews of systematic reviews and meta-analyses, indicating no effect of avocado consumption on glucose level on the one hand, but a significant impact on reductions in fasting insulin and HbA1c levels on the other, among overweight and diabetic populations [26]. It is essential to note, however, that most studies have focused on the consumption of the whole fresh avocado fruit rather than isolated bioactive compounds; further research is needed to evaluate the mechanisms by which avocado regulates glucose metabolism.

### 3.3. The Comparison with Antidiabetic Medications

In the present paper, the glucose-lowering efficacy of selected plant-derived substances was evaluated to compare their effects against metformin and other antidiabetic agents used in T2D management. No head-to-head studies directly comparing MH with metformin were identified. Nevertheless, data for *Persea americana* permit an indirect appraisal of hypoglycemic potential. Kouamé et al. reported that 28-day administration of a methanolic leaf extract of *Persea americana* (100 mg/kg body weight) reduced blood glucose in T2D rats by 37.4%. In that model, the pharmacological comparator was glibenclamide, as well as metformin (used in the intestinal glucose absorption assay). The post-treatment glucose concentrations were 145 mg/dL for the extract versus 133.8 mg/dL for glibenclamide. These results showed that the extracts of *Persea americana* have comparable effects to metformin by inhibiting postprandial hyperglycemia, as well as antihyperglycemic effects similar to glibenclamide [27]. Consistent findings revealed that an AvX at the dose of 26.7–106.6 mg/kg lowered fasting glucose in diabetic rats from approximately 400 mg/dL (diabetic control) to 85.7 mg/dL after 14 days (≈79% reduction), while metformin yielded a final value of 88.4 mg/dL. Although these end-points were numerically similar, a direct statistical comparison between the highest AvX dose and metformin was not provided [28]. Moreover, in other observations in rats maintained on a high-sugar diet, metformin increased glycemia to 123.75 mg/dL (*p* < 0.01) and augmented body-weight gain (125.7%) vs. controls (114.4%). In contrast, an infusion prepared from AvX maintained glucose levels near control values (85.75 mg/dL) and attenuated weight gain (94.3%), indicating a more favorable metabolic response to the extract [29]. These findings support pronounced hypoglycemic and antioxidant actions of *Persea americana* leaf and seed preparations. The magnitude of glucose reduction, accompanied by improvements in insulin sensitivity, appears in some models to approximate effects seen with standard antidiabetic therapy, including metformin; however, in the absence of long-term, human head-to-head comparisons, such inferences should be regarded as indirect. It should be noted that the datasets discussed above refer primarily to extracts obtained from the leaves and seeds of *Persea americana*. Their hypoglycemic activity results from the presence of a complex array of bioactive compounds, including polyphenols, flavonoids, terpenoids, and tannins. Consequently, the pharmacodynamic equivalence observed in selected experimental models when compared with conventional antidiabetic drugs such as glibenclamide or metformin reflects the multifaceted, synergistic actions of the complete plant extracts rather than the effect of any single isolated constituent.

### 3.4. Mechanism of Action

In mammals, hexokinase (HK) exists in four isoforms, HK-1, HK-2, HK-3, and glucokinase, which exhibit different tissue distributions, perform specific physiological functions, and possess unique kinetic properties depending on substrate availability and environmental conditions. The isoenzymes HK-1, HK-2, and HK-3 are characterized by a high affinity for glucose, as they are effective even at concentrations below 1 mM. Their activity is strongly regulated by the reaction product, glucose-6-phosphate (G6Pase). MH modulates glucose metabolism by inhibiting the enzymes initiating glycolysis, primarily HK-1 and HK-2, as well as glucokinase. HK catalyzes the first step of glycolysis, the phosphorylation of glucose to G6Pase. MH acts as a competitive inhibitor, competing with glucose for the active site of the enzymes, and exhibits non-competitive action by altering their conformation, which further reduces enzymatic activity. Additionally, MH has been associated with an increase in hepatic gluconeogenesis, reflecting a shift toward endogenous glucose production when glycolytic pathways are inhibited (Figure 1) [8]. In pancreatic β-cells, glucokinase plays a crucial role in sensing glucose concentration changes. By catalyzing the phosphorylation of glucose, it contributes to an increase in the ATP/ADP ratio, closure of ATP-sensitive potassium channels (K-ATP), influx of calcium ions (Ca^2+^) into the cytoplasm, and insulin secretion. Inhibition of glucokinase by MH leads to a slowdown of glucose metabolism in β-cells, resulting in reduced ATP production. Consequently, ATP-dependent potassium channels remain open, limiting Ca^2+^ influx into the cells, which leads to a decrease in intracellular calcium concentration and reduced insulin secretion. Moreover, the presence of MH causes a rightward shift in the glucose-insulin secretion curve, meaning higher glucose concentrations are required to trigger a secretory response [30].

It has been shown that inhibition of insulin secretion by using MH significantly affected metabolism in both laboratory rats and sand rats. In *Psammomys obesus*, which preferentially metabolizes carbohydrates during fasting while minimizing lipid utilization, administration of MH induced a significant shift toward fat oxidation. An increase in oxygen consumption by about 30%, CO_2_ production by 10%, and a decrease in the respiratory quotient (RQ) were observed, indicating enhanced lipolytic processes. Hyperglycemia, resulting from impaired glucose utilization, persisted for at least five hours, promoting a metabolic adaptation toward increased fat utilization, characteristic of prolonged fasting conditions. Similar changes were observed in laboratory rats, although the process occurred more rapidly due to faster elimination of MH from the organism (within 2–3 h), after which metabolic parameters returned to baseline values [31]. These results confirm the key role of MH in regulating insulin secretion.

### 3.5. Conclusions and Future Directions

The available data suggest that the use of low doses of MH primarily leads to a reduction in plasma insulin concentration with long-term supplementation without a clear effect on blood glucose levels. With increased doses of MH, additional metabolic effects are observed. In the long term, the use of higher doses of MH is associated with reduced blood glucose levels. These observations underscore the potential for modulating glycemic control of MH. However, it should be emphasized that this conclusion is based mainly on animal models, as comparable long-term data in humans are currently lacking. Furthermore, the review should clearly distinguish the long-term effectiveness of MH depending on the species, as current evidence demonstrates that metabolic responses to MH may differ substantially between animal models and humans. This distinction is crucial for accurately interpreting existing findings and for determining the translational relevance of animal-based results to human therapy. It is essential to fully evaluate the mechanisms of MH action to precisely determine the optimal therapeutic doses and the duration of therapy for the improvement of tissue insulin sensitivity. Moreover, not only MH participate in carbohydrate metabolism improvement, but also whole avocado consumption seems to have a beneficial effect on glucose regulation.

## 4. β-Carotene

### 4.1. General Characterization

β-Carotene belongs to the group of naturally occurring carotenoid pigments primarily synthesized by plants that are responsible for the characteristic bright yellow, orange, and red coloration of fruits and vegetables. This compound is also found in certain species of fungi, algae, and phototrophic bacteria, which are photosynthetic organisms. As the human body cannot endogenously synthesize β-carotene, it must be obtained exogenously through diet or supplementation [32,33]. It belongs to terpenoids (isoprenoids), which are synthesized biochemically from eight isoprene units; thus, it possesses 40 carbons [34]. Carotenoids, including β-carotene, fulfill several critical biological functions and are considered among the most important bioactive components of the human diet. Their role in the prevention of non-communicable diseases, particularly metabolic disorders such as T2D and obesity, has been the focus of intense scientific investigation. β-carotene exhibits potent antioxidant properties, allowing it to neutralize reactive oxygen species (ROS), thereby mitigating oxidative stress and preserving cellular integrity. This mechanism may be particularly relevant in the context of preventing IR and the development of T2D [35,36]. Moreover, β-carotene serves as a provitamin A compound. Its metabolites, such as retinal and retinoic acid, exert hormone-like effects and participate in the regulation of numerous physiological processes [35]. The significance of β-carotene is further supported by population-based dietary observations. Eating patterns considered beneficial for metabolic health, such as the Mediterranean diet and the Dietary Approaches to Stop Hypertension (DASH) diet, are characterized by high intake of carotenoids, including β-carotene. This suggests that regular consumption of carotenoid-rich foods may contribute to the prevention of metabolic diseases, particularly in the context of an aging population and the growing global obesity epidemic [37].

### 4.2. The Effect on Glucose and Insulin Levels

#### 4.2.1. In Vitro Studies

β-Carotene has shown promising insulin-like activity in cellular models of insulin resistance. In hepatic HepG2 cells, which respond more weakly to insulin compared to primary hepatocytes, treatment with β-carotene at concentrations of 10–20 μM led to a marked enhancement of glucose uptake, reaching approximately a 1.5-fold increase compared with untreated controls. A similar effect was observed in L6 skeletal muscle cells, where glucose uptake rose by about 1.4-fold, closely mirroring the response to insulin stimulation (1.49-fold). Notably, these effects occurred with minimal cytotoxicity, indicating that β-carotene can safely promote cellular glucose utilization. Although the underlying molecular mechanisms were not fully delineated in this study, the observed outcomes are consistent with pathways typically involved in insulin signaling, suggesting a potential role for β-carotene in improving glucose handling in IR tissues [38]. Recent studies have extended these findings by uncovering synergistic actions and gene-regulatory mechanisms of β-carotene in vitro. Co-treating human adipocytes with β-carotene and the citrus flavonoid naringenin markedly upregulated thermogenic and metabolic genes (UCP1—uncoupling protein 1; GLUT4, adiponectin) and elevated protein levels of PPARα, PPARγ, and PGC-1α—key regulators of lipid oxidation and insulin sensitivity [39]. Moreover, studies using an insulin-resistant gestational diabetes model demonstrated that β-carotene elevates sex hormone binding globulin (SHBG) expression, which subsequently enhances GLUT4 expression, leading to improved glucose uptake and reduced insulin resistance [40]. SHBG serum level is strongly correlated with T2D [41]. Thus, in vitro evidence underscores the potential of β-carotene to restore insulin signaling and glucose uptake across multiple tissues and pathways (Table 2).

#### 4.2.2. Animal Studies

In vivo, β-carotene has shown significant antidiabetic effects in animal models of hyperglycemia. In HF diet-induced obese mice, low-dose β-carotene supplementation (3 mg/kg/day) improved insulin sensitivity and reduced hepatic fat accumulation. Notably, combining β-carotene with the antidiabetic drug metformin produced a synergistic benefit—together they enhanced fatty acid oxidation and glucose utilization more than either treatment alone, partly by upregulating genes involved in fatty-acid uptake and glucose transport in muscle and liver [42]. Similarly, in aged diabetic rats, a multi-nutrient regimen of β-carotene together with magnesium and zinc, metformin resulted in superior glycemic control and insulin sensitivity compared to metformin monotherapy [43].

β-carotene may aid glycemic control through incretin stimulation. A recent study showed that feeding carotenoid-rich *Momordica cochinchinensis* (Gac) fruit aril, which is high in β-carotene, to T2D mice significantly improved FBG, glucose tolerance, and insulin sensitivity. These metabolic benefits were accompanied by a ~2-fold increase in circulating GLP-1 levels and enhanced β-cell function, effects largely attributed to β-carotene’s ability to activate intestinal sweet taste receptors and stimulate GLP-1 secretion. Notably, the glycemic improvements were attenuated in GLP-1 receptor–knockout mice, confirming that the antidiabetic action of Gac aril (and its β-carotene content) is mediated in part through the incretin pathway [44]. Animal studies show that β-carotene can improve glycemic control via multiple mechanisms—by enhancing peripheral insulin sensitivity, supporting metabolic gene expression, and augmenting incretin hormones, which highlights its potential as a natural adjunct for managing carbohydrate disorders (Table 3).

#### 4.2.3. Human Studies

Canas et al. evaluated the efficacy of a six-month dietary intervention involving supplementation with a fruit and vegetable juice concentrate (FVJC) rich in β-Carotene, in the context of IR and excess body weight. The study group consisted of 30 prepubertal boys (n = 21 with overweight; n = 9 with normal weight). At baseline, according to Homeostatic Model Assessment for Insulin Resistance (HOMA-IR) index, differences in triglycerides, pro-inflammatory markers (hs-CRP, IL-6), and the leptin-to-adiponectin ratio have been shown (with worse values in the group with overweight). Furthermore, it was demonstrated that serum β-Carotene concentration was inversely proportional to HOMA-IR and abdominal fat content. Following the intervention, a statistically significant increase in serum β-carotene concentration was observed in the group receiving FVJC (approximately 303% in lean boys and 334% in boys with overweight). The six-month study period demonstrated that FVJC supplementation significantly improved glucose homeostasis in boys with overweight, with a significant reduction in the Log HOMA-IR index (from 1.268 to 1.204) vs. placebo (increase in HOMA-IR from 1.339 to 1.429). Furthermore, the intervention also had a beneficial effect on the body composition of the participants with overweight. Alongside a significant reduction in triglyceride levels, a 1.47% reduction in visceral adipose tissue mass was observed [45].

In a subsequent study, Asemi et al. analyzed the metabolic, anti-inflammatory, and antioxidant benefits of consuming synbiotic food supplemented with β-carotene in patients with T2D (n = 51). Patients in the intervention group received a synbiotic preparation enriched with β-carotene. The daily dose of the product contained the probiotic *Lactobacillus sporogenes* (1 × 10^7^ CFU), the prebiotic inulin (0.1 g), and β-carotene (0.05 g). The entire study was initiated with a two-week run-in period, aimed at preparing participants by eliminating all other probiotic or synbiotic products from their diet. This stage was followed by the main six-week intervention phase, during which participants consumed their randomly assigned preparation, either the synbiotic or a placebo. Subsequently, a three-week “washout” period was introduced to return the body to its baseline state. In the final stage, also lasting six weeks, a crossover model was employed, involving a switch of preparations between the groups individuals who initially received the synbiotic began taking the placebo, and vice versa. At the end, after 6 weeks of intervention, no statistically significant difference in the change in fasting plasma glucose (FPG) was found between the group receiving the synbiotic and the placebo group (*p* = 0.50). However, a significant reduction in insulin concentration was recorded compared to the placebo (mean change: −1.00 ± 7.90 vs. +3.68 ± 6.91 µIU/mL; *p* = 0.002). This change translated into a significant improvement in the HOMA-IR index (mean change: −0.73 ± 3.96 vs. +1.82 ± 4.09; *p* = 0.002). A significant difference between the groups was also demonstrated for the Homeostatic Model Assessment for β-cell function (HOMA-B) index (*p* = 0.01). The results of this study corroborate previous reports, suggesting a beneficial effect of β-carotene supplementation on the lipid profile [46].

A comparison of study results in both pediatric and adult populations suggests that β-carotene supplementation may beneficially affect glucose-insulin homeostasis. However, it must be emphasized that in both analyzed cases, β-carotene was administered as a component of a complex food matrix—a synbiotic and a fruit-and-vegetable juice, respectively. This gives rise to the key hypothesis that the observed metabolic effects may not result from the action of the carotenoid itself, but rather from its synergistic interaction with other bioactive components of the product. Although a meta-analysis of Jiang et al. showed that both dietary and circulating concentrations of β-carotene were associated with a lower risk of T2D (RR—relative risk: 0.78, 95% CI—confidence interval: 0.70 to 0.87; and RR: 0.60, 95% CI: 0.46 to 0.78, respectively) [47]. Thereby, more clinical studies are needed to confirm the role of foods rich in carotenoids in the prevention and treatment of T2D.

### 4.3. The Comparison with Antidiabetic Medications

The comparison between β-carotene with antidiabetic medications was conducted mostly on animal models. In a streptozotocin-induced diabetic rat model, it was demonstrated that after 21 days of treatment, metformin (850 mg/70 kg b.w.) reduced blood glucose levels to 7.48 ± 0.69 mmol/L, and β-carotene (10 mg/70 kg b.w.) to 13.88 ± 0.33 mmol/L, respectively. The combined regimen (metformin 425 mg/70 kg + β-carotene 5 mg/70 kg) further decreased glycemia to 5.82 ± 0.31 mmol/L, approximating values observed in normoglycemic controls (~5.7 mmol/L). All reductions were statistically significant compared with diabetic controls (*p* < 0.001), and the combined therapy exhibited superior efficacy relative to either monotherapy [48]. Consistently, in a four-week HFD mouse model, it was reported that β-carotene supplementation (3 mg/kg/day) significantly decreased FBG, whereas metformin alone at 100 mg/kg/day did not significantly influence this parameter. The most pronounced hypoglycemic response was observed with the combined β-carotene + metformin treatment, underscoring the enhanced effectiveness of the dual intervention [42]. Collectively, these findings indicate that both agents independently exert antihyperglycemic effects, while their co-administration achieves the most substantial normalization of fasting glycemia, approaching normoglycemic levels in the streptozotocin model and yielding the strongest outcome in the HFD model [42,48]. Moreover, animal studies indicated that the use of β-carotene improves glucose metabolism with no toxic effect on hepatic cells [38]. On the contrary, it is well established that a vitamin A overdose may have negative health consequences [49]. To sum up, there is a need to establish the doses and the time of use of β-carotene, especially in the human population.

### 4.4. Mechanisms of Action

Cumulative evidence from experimental models highlights the potential of β-carotene as a bioactive compound capable of improving glycemic control through multiple mechanisms. In vitro, β-carotene enhances glucose uptake by promoting PI3K/Akt signaling, GLUT4 translocation, and adipocyte thermogenic gene expression, while also reducing oxidative stress and IR in hepatocytes and skeletal muscle cells [50,51,52].

The cellular mechanism of β-carotene has been demonstrated in HTR-8/SVneo cells (immortalized human trophoblast cells derived from placental tissue, modified by viral transfection to enable long-term proliferation in culture) [40]. Administration of β-carotene resulted in a significant upregulation of SHBG. This upregulation was associated with increased expression and membrane translocation of glucose transporters GLUT4 and GLUT3, thereby enhancing glucose uptake capacity. β-carotene treatment also led to enhanced expression of insulin receptor substrate genes IRS1 and IRS2, correlating with increased activation of PI3K. Consequently, phosphorylation of Akt was intensified, promoting the translocation of GLUT4 to the plasma membrane and its subsequent internalization, which facilitates glucose uptake into the cell [40].

Furthermore, a combination of β-carotene and naringenin activates nuclear receptors PPARα/γ, while carotenoids act as ligands for retinoid X receptors (RXR). This dual activation reprogrammed adipocytes in IR individuals, leading to increased expression of UCP1 and again GLUT4. Additionally, increased activity of triglyceride lipases and elevated glycerol release from cells were observed, indicating enhanced lipolytic activity [39].

### 4.5. Conclusions and Future Directions

In preclinical studies (in vitro and animal trials), β-carotene supplementation has been shown to improve FBG, insulin sensitivity, and β-cell function. Synergistic effects have been observed when β-carotene is combined with standard antidiabetic therapies such as metformin or trace minerals [43,50]. Furthermore, β-carotene-rich fruit extracts appear to stimulate GLP-1 secretion via activation of intestinal sweet taste receptors, thereby supporting glucose homeostasis through incretin pathways [44]. These pleiotropic actions suggest that β-carotene could serve as a valuable adjunct in the dietary or pharmacological management of IR and hyperglycemia. Nevertheless, further clinical trials are warranted to validate its translational potential and determine optimal dosing strategies.

## 5. Resveratrol

### 5.1. General Characterization

Resveratrol (3,4,5′-trihydroxystilbene) is a stilbenoid natural polyphenol commonly found in the skin of red grapes, wine, coffee, blueberries, raspberries, and peanuts. In the last few decades, resveratrol has obtained significant scientific interest due to its remarkable biological activities. Years of studies have shown its antioxidant, anti-inflammatory, antimicrobial, and potential anti-aging properties [53,54,55]. These results were obtained both during in vitro and in vivo trials [56,57]. It is considered that a dose of 5 g/day has no toxic effects and can be safely used in human supplementation for 1 month [58]. In recent studies, it has demonstrated a promising effect in the treatment of T2D and obesity due to its anti-inflammatory and glucose metabolism-enhancing properties [56]. The above-mentioned effects have been linked to various health benefits, including cardiovascular and neuroprotection [59,60].

Resveratrol occurs in two isomeric forms, trans- and cis-, with the trans-isomer being more stable and biologically active. It belongs to the class of non-flavonoid polyphenols and is composed of two aromatic rings linked by an ethylene bridge. The presence of hydroxyl groups at specific positions contributes to its antioxidant potential. Natural dietary sources typically provide small amounts of resveratrol, with higher concentrations found in red wine and *Polygonum cuspidatum* root extracts [61]. Although resveratrol is well absorbed in the intestine, its systemic bioavailability is limited (<1%) due to rapid metabolism via glucuronidation and sulfation in the liver and gut. As a result, only trace amounts of the parent compound reach the circulation, while the majority is present in conjugated forms. Nonetheless, emerging studies suggest that these metabolites may retain some biological activity [62,63].

Resveratrol has been evaluated in multiple preclinical and clinical studies for its potential protective effects in metabolic disorders. While the compound is generally well-tolerated and safe at moderate to high doses, its therapeutic application remains limited by pharmacokinetic barriers. Recent developments in pharmaceutical technology, such as nanoformulations and improved delivery systems, aim to enhance its oral availability and broaden its clinical utility [64].

### 5.2. The Effect on Glucose and Insulin Levels

#### 5.2.1. In Vitro Studies

An in vitro study has shown different effects of resveratrol on human cells, depending on the subject’s BMI. The study consisted of muscle biopsies from 8 lean women (BMI = 21.9 ± 0.7 kg/m^2^) and 8 women with severe obesity (BMI 46.1 ± 3.1 kg/m^2^). Myotubes were then incubated for 24 h in 0.025% dimethyl sulfoxide (DMSO) containing 1 µM of resveratrol dissolved. There was an increase in AMPK expression in both groups, and in SIRT1 expression only in the group with lean subjects. Furthermore, glucose oxidation in lean individuals marked an increase [65]. It shows that resveratrol supplementation could lower glucose levels in vivo and maintain high glucose breakdown during energy-intensive exercises.

Another in vitro study conducted on HepG2 liver cells analyzed the effect of resveratrol and high glucose concentration (HG—40 mM) on the expression of SIRT1 protein, which affects glucose metabolism in the liver. The cells used in the study were cultured for 48 and 72 h under various conditions: without resveratrol (control), with a low concentration of resveratrol (LR—25 µM), with a high concentration of resveratrol (HR—50 µM), with a HG, as well as with a combination of HG and LR (HG+LR), and HG and HR (HG+HR) concentrations. When cells were exposed to LR and HR, the SIRT1 level showed a significant increase in both 48 and 72 h. However, regardless of the elevation in the level of SIRT1 expression in LR and HR groups after 48 h, the level of SIRT1 in LR cells after 72 h decreased slightly. Cells cultured under the HG exhibited a decrease in SIRT1 expression compared to the control group. When cells were exposed to HG+LR and HG+HR, the level of SIRT1 expression was even greater in comparison to cells cultured without an elevated level of glucose [57]. These results may indicate that the use of resveratrol can produce effects related to glycemic control, particularly in individuals with diabetes who have elevated blood glucose levels.

#### 5.2.2. Animal Studies

A separate study investigated the effects of resveratrol on glucose metabolism and insulin sensitivity in murine models. In a diet-induced obesity model, mice were fed a HF diet or a standard chow diet. Although resveratrol administration did not alter fasting glucose levels, it significantly reduced fasting insulin levels. A hyperinsulinemic-euglycemic clamp study revealed that HF-fed mice exhibited a decreased glucose infusion rate (GIR) compared to chow-fed animals. Importantly, resveratrol-treated HF mice demonstrated a significantly higher GIR relative to HF controls, indicating improved insulin sensitivity. Furthermore, in a genetic model of diabesity using diabetic mice (KKAy strain) maintained on an HF diet, treatment with resveratrol (400 mg/day/kg for 10 weeks) significantly enhanced glucose tolerance, as evidenced by an OGTT (2 g glucose/kg), and reduced fasting glucose levels [66].

Another study examined the effects of resveratrol on rats with gestational diabetes mellitus (GDM) over two weeks. The study compared FBG levels and blood insulin concentrations. A total of 100 rats with GDM were randomly divided into five groups: one control group without supplementation, three groups receiving resveratrol at doses of 60, 120, and 240 mg/kg, and one group treated with metformin at 200 mg/kg. Fasting glucose levels decreased in all treatment groups; however, only the group receiving 240 mg/kg of resveratrol achieved glucose levels comparable to those observed with metformin treatment—the standard therapy for diabetes (9.7 ± 1.5 mmol/L in the 240 mg/kg resveratrol group vs. 8.8 ± 1.3 mmol/L in the metformin group). Regarding insulin levels, the 120 mg/kg resveratrol group showed values comparable to the metformin group, while the 240 mg/kg resveratrol group exhibited even better results—similar to those of healthy rats [67]. These findings suggest that resveratrol supplementation in humans may have a beneficial effect on FBG and insulin levels in the context of GDM.

#### 5.2.3. Human Studies

A clinical study was conducted on 45 subjects, both male and female, aged between 30 and 70 years, who were diagnosed with T2D and undergoing hypoglycemic treatment, with an HbA1c level of 7 or higher, and not obese due to a BMI less than 35 kg/m^2^. 25 participants were given 400 mg capsules of 99% pure trans-resveratrol, which had to be taken twice a day for a period of 8 weeks. Other participants were given 400 mg of placebo capsules with the same supplementation recommendations. The study showed a significant decrease in FBG in the resveratrol group (−31.84 ± 47.6 mg/dL), although HbA1c levels did not differ between the groups [53].

A study conducted by Poulsen et al. enrolled 24 males aged 18–70 years with a BMI higher than 30 kg/m^2^. One group was given 500 mg trans-resveratrol tablets, and the other was given a placebo. Both groups were instructed to take 1 tablet thrice a day for a period of 4 weeks. Blood results showed no noteworthy changes in the level of blood glucose or the HbA1c count [68].

A major clinical study involved 472 elderly patients (>60 years old) diagnosed with T2D. One group of 242 subjects was given 500 mg/day of resveratrol for 6 months. The other group (n = 230) was given a placebo also for 6 months. Blood samples were taken every 2 months. The first significant changes in blood test results could be observed after 4 months of therapy. After 6 months, there were major changes in the level of HbA1c, SIRT1, and AMPK expression and G6Pase activity in the resveratrol group. HbA1c count dropped over 2%. SIRT1 and AMPK expression improved, and G6Pase activity was greater [56].

A meta-analysis included 15 different clinical trials, involving 921 participants, divided into two age groups: 45–59 years old and 60 years old or older. It showed that there was a correlation between age and dose of resveratrol, and its blood glucose-lowering properties. Best results on glycemic control in both age groups were observed when subjects were administered doses of 250–500 mg of resveratrol per day. Glycemic control includes HbA1c level, blood insulin level, and glucose concentration, which were significantly decreased. Additionally, elderly patients (≥60 years old) showed a significant improvement in insulin sensitivity, as indicated by a reduction in the HOMA-IR [69]. Other meta-analyses on 871 T2D patients confirmed these results, showing superiority of resveratrol to placebo on fasting blood glucose with doses ≥ 500 mg (MD: −13.34; 95% CI: −22.73 to 3.95; *p* = 0.005) and HbA1c at three months of use (MD: −0.41; 95% CI: −0.65 to −0.16; *p* = 0.001) (Table 4) [70]. It is worth noting that trials by Paulsen et al. and Khodabandehloo et al. are both included in meta-analyses by García-Martínez et al. Nevertheless, taking into account the above parameters, it seems that resveratrol beneficially modulates glycemic control.

### 5.3. The Comparison with Antidiabetic Medications

Resveratrol (93.68 ± 3.51 mg/kg/day) shows moderate but inconsistent efficacy compared to metformin (231.28 ± 12.24 mg/kg/day) in improving glucose homeostasis and insulin sensitivity in mice. It is worth noting that the glucose area under the curve was reduced by ~10% and ~20%, respectively, in resveratrol and metformin-treated mice. Resveratrol appears to be more effective at preventing diet-induced metabolic derangements and reducing adipocyte size in obese diabetic mice, while metformin produces stronger effects in treating established IR and improving insulin tolerance. Notably, combining metformin and resveratrol showed an additive effect on glucose tolerance [71]. Resveratrol treatment, alone (20 mg/kg/day) or in combination with metformin (150 mg/kg/day), resulted in a significant improvement in glucose and other metabolic parameters, such as triglycerides and cholesterol levels. Additionally, resveratrol reduced liver damage and improved renal function in diabetic mice compared to metformin [72]. In a meta-analysis by García-Martínez et al., containing studies performed on patients with T2DM, the Effect Size of resveratrol on glucose was −13.36 mg/dL (*p* = 0.0007) with a duration < 3 months, on insulin −0.94 mIU/L (*p* = 0.007), on HbA1c −0.22% (*p* = 0.02), and on HOMA-IR −0.83 (*p* = 0.04) [73]. These findings suggest that resveratrol may serve best as an adjunct to metformin rather than as a standalone replacement. Although resveratrol is usually well-tolerated, it needs to be concluded that its high doses may have a potentially toxic effect [74]; thus, more extensive research is warranted in order to validate the current findings.

### 5.4. Mechanisms of Action

The primary effect of resveratrol in glucose metabolism is the stimulation of AMPK expression along with an enhancement of SIRT1 and PGC-1α activity [65,66]. Resveratrol inhibits the mitochondrial F1F0–ATPase/ATP synthase, leading to an increase in the adenosine monophosphate (AMP) level necessary for AMPK activation [75,76,77]. AMPK is mainly responsible for mitochondrial biogenesis and proper energy homeostasis, which directly translates into glucose requirement. Its activation also results in suppressed lipogenesis and glycogenolysis while promoting fatty acid oxidation, glucose uptake by GLUT4, and glycolysis [75]. The SIRT1 protein consists of a catalytic core and two domains, the C-terminal and N-terminal regions [78]. Resveratrol binds to both of these domains, enhancing SIRT1 activity by interacting with peptides such as p53-7-amino-4-methylcoumarin (P53-AMC). Moreover, resveratrol facilitates phosphorylation within the C-terminal domain by strengthening the interaction between SIRT1 and liver kinase B1 (LKB1), another major element involved in AMPK activation [55,79]. PGC-1α, activated through SIRT1-dependent deacetylation, serves as a key regulatory factor in several metabolic pathways, initiating gluconeogenesis and fatty acid β-oxidation through nuclear hormone receptors such as PPARα, Estrogen-related receptor alpha (ERRα), and Hepatocyte Nuclear Factor 4 alpha (HNF4α) [80]. Through the coordinated activation of AMPK, SIRT1, and PGC-1α, resveratrol enhances cellular energy efficiency and metabolic flexibility. These effects collectively contribute to improved glucose homeostasis and protection against metabolic disorders.

### 5.5. Conclusions and Future Directions

Resveratrol has been shown to enhance glucose metabolism by increasing the expression of critical proteins like SIRT1 and AMPK, improving insulin sensitivity, and boosting glucose oxidation [81]. In vitro and animal trials both offered promising insights into resveratrol treatment. Human clinical trials also revealed that resveratrol supplementation has beneficial effects on the human body. However, its effectiveness depends on variables such as dosage, treatment duration, and BMI. The greatest results were discovered after a resveratrol treatment period exceeding 1 month [53,56]. Four weeks of treatment did not show any change in glucose metabolism [68]. Moreover, the response to resveratrol was weaker in patients with obesity [65].

In conclusion, resveratrol has proven that its properties can be useful in maintaining proper blood glucose levels, but knowledge about it is still insufficient, prompting further research to consider resveratrol therapy in the future.

## 6. Steviosides

### 6.1. General Characterization

*Stevia rebaudiana* is a plant that occurs naturally in South America, particularly in Paraguay and Brazil [82]. It is currently the subject of numerous studies regarding glycemic control and health properties. It is 250–300 times sweeter than sucrose. *Stevia rebaudiana* contains more than 30 different steviol glycosides, among which stevioside and rebaudioside A are the most popular. Stevioside, a diterpene glycoside, contains three molecules of glucose and a glucone moiety, known as steviol. The same structure (diterpene glycoside) possesses rebaudioside A. Other compounds isolated from *Stevia rebaudiana* are several types of rebaudiosides (from A to F, M, U, T, R, and S), steviolmonoside, steviolbioside, rubusoside, dulcoside, and A. Most of the isolated diterpenoid glycosides present an analogous chemical backbone structure (steviol), differing in the context of carbohydrate at positions C13 and C19. *Stevia rebaudiana* also contains steviol and isosteviol (aglykons), which are products of the hydrolysis of stevioside [83]. Most of them can be extracted from the plant’s leaves [84]. Stevia was legally introduced to the European Union in November 2011, as a food additive under the designation E960 [85]. The substance is approved for use in products such as breakfast cereals, cereal bars, certain confectionery products, ice cream, yogurts, and flavored milk [86].

Recent studies aim to utilize in vitro culture techniques in plant cultivation to develop varieties with an improved rebaudioside A-to-stevioside ratio. This approach is driven by enhancing the sweetener’s organoleptic properties, as stevioside is responsible for its bitter aftertaste [87]. The calorie content of stevia is 0 kcal per 100 g of the product. This zero-calorie nature of the substance directly influences overall daily caloric intake. This characteristic is particularly beneficial for individuals with impaired glycemic control, as it does not contribute to postprandial glucose elevation. Reduced sugar intake through the use of stevia at a dose of 4 mg/kg body weight, which is the maximum allowed dose in Europe, not only lowers cholesterol, triglycerides, LDL, and HDL, but also reduces HbA1c [88]. Weight loss and stevioside consumption have been shown to reduce the secretion of pro-inflammatory molecules in adipose tissue, including tumor necrosis factor alpha (TNF-α), interleukin-6 (IL-6), interleukin-10 (IL-10), interleukin-1β (IL-1β), keratinocyte-derived chemokine (KC), and macrophage inflammatory protein-1 alpha (MIP-1α). These cytokines initiate the inflammatory response by engaging cells of the immune system. Moreover, steviosides appear to decrease the activation of the nuclear factor kappa B (NF-κB) signaling pathway, which plays a central role in regulating inflammatory responses [89]. Antioxidant properties of *Stevia rebaudiana* have also been observed. It contains many phenolic compounds, able to neutralize free radicals, thus preventing different metabolic conditions [90]. Rebaudioside A has been demonstrated to have many physiological functions in preventing and treating not only diabetes, but also in hypertension or some cancers (e.g., colon cancer) [91,92].

### 6.2. The Effect on Glucose and Insulin Levels

#### 6.2.1. In Vitro Studies

In isolated mouse (in vitro) pancreatic islets maintained under both static incubation and dynamic perifusion to emulate physiological secretory patterns, rebaudioside A (10^−16^–10^−6^ mol/L) produced a clear, dose-dependent enhancement of insulin release, but only when extracellular glucose concentrations were elevated to ≥8.3 mmol/L. At basal or low glucose levels (3.3–5.5 mmol/L), it remained inactive, demonstrating a strictly glucose-dependent insulinotropic profile. Removal of extracellular Ca^2+^ completely abolished this effect, confirming that rebaudioside A requires voltage-dependent calcium influx to trigger insulin granule exocytosis. Moreover, pharmacological opening of ATP-sensitive K^+^ channels with diazoxide hyperpolarized β-cells and suppressed insulin secretion; rebaudioside A was unable to overcome this inhibition, indicating that its mechanism does not involve direct closure of K-ATP channels [93]. Collectively, these findings identify rebaudioside A as a potent, glucose-dependent secretagogue with a low risk of hypoglycemia, warranting further investigation into its downstream signaling targets.

In a study conducted on rat intestinal tissue, the effects of both steviol (as a product of enzymatic hydrolysis of stevioside) and stevioside at concentrations of 1 mM and 5 mM, respectively, on glucose absorption were evaluated. The results demonstrated that steviol at a concentration of 1 mM reduced glucose absorption by 29%. This effect was attributed to decreased ATP levels resulting from the inhibition of mitochondrial NADH–cytochrome c reductase and cytochrome oxidase activity, as well as alterations in the metabolism of intestinal absorptive cells. It is important to note that stevioside itself did not exhibit any activity or effect; the observed inhibition was solely due to its metabolite, steviol. Achieving such concentrations (1–5 mM) in the human intestine may be difficult following standard dietary intake of stevioside [94]. What is more, stevioside and rebaudioside A are hydrolyzed in the gastrointestinal tract after oral intake, which leads to the formation of steviol. The latter one is absorbed into the circulation and is at least eliminated mainly as steviol glucuronide via excretion into the urine [95].

In the next study conducted on rat-derived cell lines—L6 myoblasts and 3T3-L1 adipocytes, it was demonstrated that administration of stevioside (a diterpene glycoside extracted from *Stevia rebaudiana*) significantly upregulated the expression of the GLUT4 gene. This led to an increased abundance of GLUT4 proteins within the cell membrane, thereby enhancing the cells’ capacity to absorb glucose and contributing to improved glucose homeostasis [96].

The efficacy of various steviol glycosides (rich in stevioside and rebaudioside A) was evaluated in primary cardiac fibroblasts isolated from neonatal rats. The compounds were administered at a concentration of 1 mg/mL for 24 h. The results demonstrated increased phosphorylation of Akt and enhanced translocation of GLUT4 transporters toward the cell membrane. The interaction of steviol glycosides with the insulin receptor was confirmed by the use of the insulin receptor antagonist S961. These findings suggest that steviol glycosides may contribute to improved glycemic control [97].

Alpha-amylase and α-glucosidase catalyze the hydrolytic cleavage of glycosidic bonds in dietary carbohydrates, yielding glucose and requiring one molecule of water per cleavage. In an in vitro study, a water-soluble stevia extract incorporated into a specially formulated bread matrix inhibited both enzymes in a concentration-dependent manner. Enzyme activity progressively decreased as the extract concentration increased from 50 to 1000 µg/mL. Such inhibition could potentially attenuate postprandial glycemic excursions in individuals with diabetes by slowing the conversion of complex carbohydrates into absorbable glucose [98].

#### 6.2.2. Animal Studies

The study conducted on rats assessed the effect of steviol glycosides on glycemic control. The experiment was carried out on 70 male Wistar rats. The majority of these rats (60 individuals) were fed an HF diet for eight weeks (with ad libitum access to food). The next step was to induce diabetes, which was achieved by intraperitoneal injection of streptozotocin. The diabetic animals were divided into six groups and fed pure stevioside/rebaudioside A at doses of 500 and 2500 mg/kg for five weeks. In this study, no hypoglycemic or insulinotropic mechanisms were observed in the treated animals. However, an increase in circulating leptin levels was noted, corresponding to reduced food intake. An improvement in the lipid profile was also observed in the experimental group, reaching approximately half the values observed in healthy control rats. These findings suggest that therapy with steviol glycosides may be an alternative strategy for the management of T2D, as it minimizes the adverse effects of impaired lipid metabolism in diabetic subjects [99].

The next study, on 80 male Wistar rats, evaluated and analyzed the effects of supplementation with stevia leaf powder, isolated polyphenols, and fiber on metabolic and physical parameters in rats with T2D induced by streptozotocin. The rats were divided into eight groups, some of which received a standard diet, while the others were fed diets enriched with 4% stevia leaf powder, isolated polyphenols, or fiber. The administration of these compounds lasted for 30 days, both before and after the induction of T2D. In the groups supplemented with stevia leaf powder and polyphenols, a significant reduction in blood glucose levels (by 36% and 64%) was observed, as well as an increase in insulin levels compared to the diabetic control group. Additionally, the supplementation contributed to improved liver and kidney function and reduced oxidative stress, further supporting the potential of these compounds as adjunctive therapy in T2D [100].

To conclude, these two studies demonstrate differing effects on glycemia and insulin secretion. Further research on stevia is needed to determine whether compounds other than steviol glycosides may be responsible for the observed outcomes.

#### 6.2.3. Human Studies

A subsequent study investigating the hypoglycemic effects of steviosides was conducted on individuals (n = 20) after collecting psychosocial, economic, medical history, and general health data. The participants were randomly assigned to either a control group or an experimental group. The experimental group received 1 g of powdered stevia leaf daily for 60 days. After the intervention period, fasting and postprandial blood glucose levels were measured. A significant decrease in glucose levels was observed in the experimental group, in contrast to the control group, which showed no notable improvement. In the experimental group, fasting glucose levels decreased from 156.61 ± 31.32 mg/dL to 123.55 ± 22.94 mg/dL, and postprandial glucose levels decreased from 225.17 ± 43.86 mg/dL to 200.60 ± 43.80 mg/dL. However, the study is not without limitations. It lacks randomization and double-blinding, which is reflected in the absence of clinical trial registration. Moreover, it was conducted on a limited number of patients. Therefore, it should be regarded as a pilot study [101].

In a controlled clinical trial aimed at evaluating the efficacy of stevioside in glycemic control, a group of unrelated individuals (n = 150), including participants with T2D (n = 40), participants with obesity (n = 60), and healthy subjects as the control group (n = 50), were enrolled. Participants received stevia at the legally approved dose of 4 mg/kg body weight for 24 weeks. In the experimental group, the substance improved glycemic control, as assessed by fasting serum insulin levels and HbA1c, and also reduced cardiometabolic risk, reflected by decreased LDL cholesterol and triglyceride levels. Interestingly, in the obese group, supplementation was associated with an increase in body weight. It needs to be added that this study should only be regarded as a preliminary reference or inspiration for future research, as its methodological limitations are considerable. The lack of randomization and the comparison between a healthy control group and an intervention group make the cohorts non-comparable. Moreover, the intervention group had an increase in caloric intake, which could have influenced the dietary outcomes. Therefore, this study represents low-certainty evidence [88].

The meta-analysis conducted by Anker et al. evaluated the effects of a natural sweetener derived from *Stevia rebaudiana* on glycemic control, blood pressure, BMI, and lipid profile parameters (total cholesterol, HDL, LDL), based on data collected from randomized clinical trials. No statistically significant evidence was found for a reduction in FBG (MD: 2.63 mg/dL; 95% CI: 7.77, 2.51; *p* = 0.32) or HbA1c levels (MD:0.00%; 95% CI: 0.24, 0.25; *p* = 0.98), although a downward trend was observed [102]. Different results revealed other meta-analyses showing significantly reduced blood glucose levels (WMD: −3.84; 95% CI: −7.15, −0.53; *p* = 0.02), especially in individuals with higher BMI, diabetes, and hypertension (Table 5) [103]. The heterogeneity of results may result from different doses, the time of use, and the form of steviol glycosides. It is worth adding that most of the studies focus on the safety of stevia as a sweetener, rather than its therapeutic potential as a pharmacological agent. Therefore, further clinical trials involving individuals with T2D are warranted.

### 6.3. The Comparison with Antidiabetic Medications

The use of various forms of steviosides as a hypoglycemic compound, in comparison with commonly available pharmacological agents such as metformin, has confirmed its antihyperglycemic activity. In a study conducted on HFD rats, it was shown that steviosides, similar to metformin, attenuate IR and improve glycemic control in diabetic gastrocnemius muscle by facilitating the expression of GLUT4 and insulin signaling molecules [104]. Moreover, the study evaluating both individual and combined administration of metformin (250 mg/kg/day) with the aqueous extract of stevia (400 mg/kg/day) demonstrated similar effects in reducing FBG levels in rats. Moreover, combined administration of stevia aqueous extract with metformin resulted in a significant decrease in HOMA-IR value (from 2.54 ± 0.09 in diabetic control to 1.46 ± 0.04; *p* < 0.001 in resveratrol + metformin group) [105]. In another study conducted on diabetic rats, administration of the powdered form of stevia leaves significantly reduced blood glucose levels in a time-dependent manner. The study evaluated its dose- and time-dependent hypoglycemic effects in comparison with glimepiride. After 21 days of treatment, higher doses of the powdered form of stevia (250 mg/kg body weight) produced a significant reduction in blood glucose (15.76%; *p* < 0.001), while glimepiride (800 µg) 40.67% [106]. The hypoglycemic effect of steviosides, compared to antidiabetic agents, was also demonstrated in another study [107]. Nevertheless, it has to be admitted that the presented studies involved only in vitro or animal models, without human populations. What is more, overall stevia intake, generally considered safe, may have potential negative outcomes in high doses, such as gastrointestinal disorders. It is also indicated that *Stevia* may modify the host gut microbiota [108]; therefore, more clinical research is warranted.

### 6.4. Mechanism of Action

Glucose is transported from the intestinal lumen into enterocytes via secondary active transport mediated by the sodium-dependent glucose cotransporter (SGLT1), located on the apical membrane of intestinal epithelial cells (Km ≈ 0.4 mmol/L). This process is driven by the inward sodium gradient maintained by the sodium–potassium ATPase (Na^+^/K^+^-ATPase) and requires adequate ATP availability [109].

Steviol decreases intracellular ATP levels in enterocytes by inhibiting the mitochondrial electron transport chain, specifically the NADH–cytochrome c reductase and cytochrome oxidase, key enzymes involved in oxidative phosphorylation. Although the function of SGLT1 and Na^+^/K^+^-ATPase remains intact, the resulting energy deficit impairs the efficiency of glucose transport [94].

Stevioside enhances glucose uptake in skeletal muscle and adipose tissue by increasing the transcription of the GLUT4 gene. The elevated expression of GLUT4 leads to a higher presence of glucose transporters in the cell membrane, which facilitates greater cellular uptake of glucose and supports improved glycemic regulation (Figure 2) [96]. The mechanism, as tested under in vitro conditions, likely involves the activation of the PI3K/Akt signaling pathway, leading to the translocation of GLUT4 to the cell membrane. To determine whether the effect of steviol glycosides mimics that of insulin, cells were treated with the insulin receptor antagonist peptide S961, which fully abolished the action of the compound, confirming that steviol glycosides may bind to the insulin receptor [97]. Additionally, a reduction in the activity of phosphoenolpyruvate carboxykinase (PEPCK), a key enzyme of gluconeogenesis, was observed [97]. Stevioside also improved glucose metabolism by activating the AMPK pathway and upregulating PPARα/carnitine palmitoyltransferase 1 [110].

### 6.5. Conclusions and Future Directions

Steviosides, natural compounds found in *Stevia rebaudiana*, have become the focus of numerous studies and scientific investigations. Due to their zero-caloric nature, natural sweeteners participate in reducing overall caloric intake, and what is more, positively contribute to glycemic control [111]. It is suggested that steviosides may enhance insulin secretion from pancreatic β-cells [112] and improve peripheral glucose sensitivity [113]. However, there is a need for large population studies evaluating these effects. Thus, more detailed studies are required to establish a specific molecular pathway of their action, particularly in individuals with diabetes [83].

## 7. Curcumin

### 7.1. General Characterization

Curcumin (diferuloylmethane) is a bioactive, low-molecular-weight polyphenol compound primarily found in the yellow-pigmented rhizome of *Curcuma longa*, commonly known as turmeric. It represents the major curcuminoid in turmeric, representing about 77% of the curcuminoid content, followed by demethoxycurcumin (17%) and bisdemethoxycurcumin (3%) [114,115]. Curcuminoids typically constitute 3–5% of the dry weight of turmeric. Curcumin has a molecular formula of C_21_H_20_O_6_ and a molecular weight of 368.37 g/mol. It is yellow-orange in color, poorly soluble in water, and more soluble in organic solvents like ethanol, acetone, and DMSO.

Curcumin exists mainly in an enolic form in solution, which contributes to its radical-scavenging and antioxidant properties. It is stable in acidic environments (pH 1–6) but degrades rapidly at neutral to basic pH, especially in aqueous media like phosphate buffer at pH 7.2. However, it shows increased stability in biological systems such as blood and cell culture media, where degradation is slower. Natural antioxidants, like ascorbic acid and glutathione, can further protect curcumin from oxidative degradation [115]. Despite its low bioavailability due to poor intestinal absorption and rapid metabolism, its bioavailability can be significantly enhanced by co-administration with agents like piperine. Curcumin is well tolerated in humans, even at high doses of up to 12 g/day.

Pharmacologically, curcumin exhibits a broad range of beneficial effects. It possesses anti-inflammatory, antioxidant, anti-obesity, anti-angiogenic, and anti-carcinogenic properties. These effects are mediated through the modulation of various molecular targets, including cytokines, transcription factors (e.g., NF-κB), enzymes, and growth factors. It has been shown to reduce inflammatory markers such as IL-1β, IL-6, TNF-α, and hs-CRP in both experimental and clinical settings [116].

In studies involving obese individuals and animal models, curcumin has been shown to improve glucose metabolism, reduce body fat, inhibit adipogenesis, and attenuate inflammation by decreasing macrophage infiltration and expression of inflammatory mediators in adipose tissue. Furthermore, it has demonstrated potential in managing metabolic disorders such as T2D and non-alcoholic fatty liver disease (NAFLD) [114].

### 7.2. The Effect on Glucose and Insulin Levels

#### 7.2.1. In Vitro Studies

Curcumin stimulated AMPK activity in rat L6 muscle cells, which was accompanied by an increase in glucose uptake. Downstream of AMPK, curcumin engaged the MEK3/6–p38 MAPK pathway, and pharmacological or genetic inhibition of either AMPK or p38 MAPK abolished the glucose uptake effect. Consistent with the cell data, curcumin given to mice elevated AMPK phosphorylation in skeletal muscle. These findings show that curcumin stimulates 2-deoxyglucose uptake and potentiates insulin-stimulated uptake, indicating its beneficial health effects [117].

Curcumin (20 μM) protected INS-1 β-cells from high-glucose/palmitate (HP; 30 mM glucose + 0.5 mM palmitate) induced dysfunction in vitro, by restoring cell viability (to 89% vs. 59% in HP alone). It significantly suppressed HP-stimulated intracellular ROS and prevented the HP-associated declines in antioxidant enzyme activities (SOD and catalase). Moreover, curcumin markedly lowered the HP-induced upregulation of several NADPH-oxidase subunits (gp91phox, p67phox, p47phox) and downregulated pro-apoptotic markers (cleaved caspase-3 and Bax). Importantly, curcumin inhibited the HP-induced decrease in cellular insulin staining, indicating preservation of islet insulin content under glucolipotoxic conditions in this in vitro model. It seems that curcumin protects β-cells from glucolipotoxic oxidative stress and preserves insulin levels, which supports the role of curcumin in glycemic control, especially by maintaining β-cell insulin-producing capacity [118].

In isolated mouse hepatocytes, curcumin suppressed hepatic glucose production in an insulin-independent manner: it inhibited gluconeogenesis from 1 mM pyruvate in a concentration-dependent way with a reported maximal decrease of 45% at 25 mM, and inhibited gluconeogenesis from dihydroxyacetone phosphate (DHAP) by 35% after 120 min exposure; curcumin also reduced glycogenolysis and potentiated insulin’s inhibitory effect on gluconeogenesis when co-administered. After 120 min of exposure to 25 mM curcumin, the decrease in key gluconeogenic enzyme activities (G6Pase and PEPCK were inhibited by ~30%, whereas FBPase was not reduced) and increased phosphorylation of AMPK (AMPK α-Thr172) were observed. Cell viability and DNA synthesis were not suppressed under these conditions, indicating that the reductions in glucose output were not due to overt hepatocyte toxicity. It can be concluded that curcumin suppresses hepatic glucose production through AMPK activation and downregulation of G6Pase/PEPCK [119].

#### 7.2.2. Animal Studies

In a streptozotocin (STZ) induced rat model of T2D, curcumin (200 mg/kg/day, given intragastrically beginning the day after STZ) partially restored serum insulin and reduced blood glucose, with immunohistochemistry showing partial recovery of insulin expression in pancreatic Langerhans islets. Curcumin treatment also improved pancreatic antioxidant status by restoring glutathione peroxidase (GPX) and superoxide dismutase (SOD) activities and by decreasing malondialdehyde (MDA) levels in pancreatic tissue homogenates, consistent with reduced oxidative stress. Histological analysis revealed a marked decrease in apoptotic cells within islets after curcumin administration, indicating preserved β-cell survival and function. These animal-study findings indicate that curcumin protects islet β-cells from STZ-induced oxidative injury and partially preserves insulin secretion [120].

In a study conducted on C57BL/6J mice fed a high-fat diet, oral curcumin supplementation (100 mg·kg^−1^·day^−1^ for 4 weeks) improved glucose tolerance, insulin sensitivity, and pyruvate tolerance and reduced hepatic triglyceride accumulation. These metabolic benefits were associated with increased insulin-stimulated Akt phosphorylation in liver, adipose tissue, and skeletal muscle and with lowered hepatic expression of key gluconeogenic genes (Pck1 and G6pc). Importantly, broad-spectrum antibiotic depletion of the gut microbiota abolished the glycemic and gene expression effects of curcumin, while fecal microbiota transplantation from curcumin-treated donors reproduced them in recipient mice, linking the metabolic actions to microbiota remodeling. Curcumin upregulated ileal Fgf15 expression and raised circulating FGF15, an effect that was also microbiota-dependent, suggesting that curcumin improves insulin sensitivity in this animal model at least in part via gut microbiota–FGF15 signaling [121].

Kato et al. showed that in orally dosed rats, highly bioavailable curcumin (theracurmin; 1.5 mg curcumin/kg, given 30 min before glucose challenge) increased portal-vein GLP-1 (total and active forms) and raised plasma insulin, producing a clear improvement in intraperitoneal glucose tolerance compared with placebo; blockade of GPR40/120 or PLC attenuated both the GLP-1 response and the glucose-lowering effect, implicating a GPR40/120–PLC pathway in the in vivo action [122].

What is important, HF diet-fed mice exhibit elevated Fgf21, but dietary curcumin (200 mg/kg/day for 12 weeks) normalized hepatic and serum Fgf21 levels while restoring FGFR1/β-Klotho and PGC-1α expression via PPARα activation. Liver-specific Fgf21 knockout mice failed to show curcumin’s metabolic benefits (improved triglyceride clearance, GTT) seen in wild-type mice, confirming FGF21’s necessity [123,124].

#### 7.2.3. Human Studies

In a 9-month, double-blind RCT (n = 240), curcumin extract (750 mg twice a day) prevented T2D development in patients with prediabetes (0% of individuals), while in the placebo group, 16.4% of subjects were diagnosed with diabetes (*p* < 0.001). Curcumin raised HOMA-β function (61.58 vs. 48.72; *p* < 0.01) and adiponectin level (61.58 vs. 48.72; *p* < 0.01) on the one hand and lowered HOMA-IR (3.22 vs. 4.04; *p* < 0.001) on the other across 3, 6, and 9 months of intervention [125].

The next study conducted by Asghari et al. in T2D patients (n = 100) co-administrated eicosapentaenoic acid—EPA (500 mg) with nano-curcumin (80 mg) for 12 weeks showed a greater reduction in insulin level [MD: −1.44 (−2.70, −0.17)], along with significant decrease in hs-CRP levels and an increase in Total Antioxidant Capacity (TAC) compared to the placebo group. These findings suggest that curcumin, together with EPA, may positively impact inflammation, oxidative stress, and metabolic parameters in patients with diabetes [126].

A 2023 meta-analysis of 59 RCTs found that curcumin/turmeric significantly lowered fasting glucose (weighted mean difference (WMD) –4.60 mg/dL; 95% confidence interval (CI) –5.55, –3.66), fasting insulin (WMD –0.87 µIU/mL; 95% CI –1.46, –0.27), HbA1c (WMD –0.32%; 95% CI –0.45, –0.19), and HOMA-IR (WMD –0.33; 95% CI –0.43, –0.22), with low heterogeneity (I^2^ < 30%) and no serious adverse effects. Subgroup analysis showed stronger effects at ≥500 mg/day doses and in studies ≥ 12 weeks [127].

An analysis of three RCTs in women with polycystic ovary syndrome—PCOS (500–1500 mg/day for 6–12 weeks) revealed significant reductions in fasting glucose (MD: −2.77, 95% CI: −4.16 to −1.38), fasting insulin (MD: −1.33, 95% CI: −2.18 to −0.49) and HOMA-IR (MD: −0.32, 95% CI: −0.52 to −0.12), alongside improved lipid profiles [128]. A similar effect was revealed in other meta-analyses [129].

Finally, an umbrella review of 22 meta-analyses confirmed curcumin’s reductions in FBG, HOMA-IR, HbA1C, and insulin, ranking it among the top nutraceuticals for glycemic control in the network meta-analysis (Table 6) [130].

### 7.3. The Comparison with Antidiabetic Medications

To assess the effectiveness of curcumin in accordance with glycemia control, Hamed et al. compared it with the use of metformin in Wistar rats. It has been shown that both curcumin and its nanoformulation act similarly to metformin and cause a decrease in IR of approximately 40%, while FBG levels decreased by 35–40% (*p* < 0.001) [131]. In another study, combined intake of curcumin and metformin resulted in greater improvement of glycemia than its single administration in rats [132]. Those results were confirmed in women with PCOS. The combination of metformin and curcumin demonstrated significant improvements in FBG level (−10.10 ± 4.87 mg/dL), insulin (−2.88 ± 2.31 mg/dL), and HOMA-IR (−0.99 ± 0.71) compared to individual agents or placebo [133]. Another study evaluated that curcumin supplementation produced a statistically significant reduction in FBG in T2D patients (MD = −11.48 mg/dL; 95%CI (−14.26 to −8.70). The pooled analysis also found a decrease in HbA1c versus placebo (MD = −0.54%; 95%CI (−0.73 to −0.35) [134]. Thus, a combination of current antidiabetic agents with curcumin seems to have a beneficial effect on glucose disturbances. It is believed that curcumin, derived from food, has low toxicity. Nonetheless, it cannot be excluded that supplementation with high doses of curcumin may cause side effects (i.a. heart, liver, kidney abnormalities) [135]. To summarize, the benefits of curcumin for human health are enormous; however, the potential challenges, such as its toxicity, bioavailability, and efficiency, need to be characterized.

### 7.4. Mechanism of Action

Curcumin activates AMPK in hepatocytes, which phosphorylates and inhibits transcription factors cAMP-response element-binding protein (CREB) and forkhead box O1 (FoxO1), leading to decreased expression of gluconeogenic genes PEPCK and G6Pase [121,136]. Concurrently, curcumin inhibits IκB kinase beta (IKKβ) and the NF-κB pathway, lowering pro-inflammatory cytokines (IL-6, TNF-α) that normally impair insulin signaling in the liver [137,138]. It also reduces cyclic adenosine monophosphate (cAMP) levels by suppressing adenylate cyclase activity, thereby diminishing CREB phosphorylation and further downregulating gluconeogenic enzyme transcription [121,139]. In rodent models, curcumin supplementation decreased hepatic PEPCK and G6Pase activity, correlating with a reduction in FBG [138,140].

Curcumin promotes IRS-1 tyrosine phosphorylation, which in turn recruits and activates PI3K, leading to phosphorylation of Akt at Ser473 and Thr308; this cascade culminates in increased GLUT4 vesicle translocation and fusion with the plasma membrane, thereby elevating glucose uptake in skeletal muscle and adipocytes [128,136]. Beyond GLUT4, activated Akt phosphorylates and inhibits glycogen synthase kinase-3 beta (GSK-3β), resulting in enhanced glycogen synthase activity and glycogen storage in muscle cells [141,142]. Curcumin also modulates mechanistic target of rapamycin complex 2, which provides an additional phosphorylation site for Akt, reinforcing insulin signaling robustness under metabolic stress [136,143]. Notably, curcumin reduces serine phosphorylation of IRS-1 by inhibiting stress kinases such as JNK and IKKβ, thus preserving IRS-1 in a signaling-competent state [140,144].

Curcumin directly stimulates intestinal L-cells to secrete GLP-1 via activation of the calcium/calmodulin-dependent protein kinase II pathway, independent of the classical cAMP/protein kinase A axis [139,145]. In parallel, curcumin inhibits dipeptidyl peptidase-4 activity—demonstrated in Caco-2 cell assays with a half-maximal inhibitory concentration (IC50) ≈ of 10 µM—thereby prolonging the half-life of endogenous GLP-1 from 1.5 to 5 min [145]. Enhanced GLP-1 levels potentiate glucose-dependent insulin secretion from pancreatic β-cells, slow gastric emptying, and inhibit glucagon release, collectively improving postprandial glycemic profiles [146,147].

Orally administered curcumin, despite low systemic bioavailability, accumulates in the intestinal lumen where it bidirectionally modulates the gut microbiome. Preclinical studies demonstrate enrichment of short-chain fatty acid (SCFA)-producing genera (e.g., *Faecalibacterium*, *Roseburia*) and *Akkermansia muciniphila*, leading to increased levels of acetate, propionate, and butyrate in the colon [137,148]. These SCFAs activate G-protein–coupled receptors FFAR2/3 on enteroendocrine cells, stimulating peptide YY and GLP-1 release, and on adipocytes/macrophages to reduce inflammation and improve insulin sensitivity [121,137]. Curcumin-induced microbiota shifts also elevate ileal FGF15 (mouse) and circulating FGF19 (human ortholog), which bind hepatic fibroblast growth factor receptor 4 to suppress gluconeogenic gene expression and lipogenesis [121,149]. Finally, restoration of gut barrier integrity—via upregulation of tight junction proteins ZO-1 and occludin—reduces metabolic endotoxemia and systemic inflammation, further enhancing insulin action [137,148].

By acting as a ligand for PPARγ in adipocytes, curcumin promotes the transcription of genes involved in adipogenesis and lipid uptake, leading to smaller, more insulin-sensitive adipocytes [139,145]. PPARγ activation by curcumin upregulates adiponectin, an adipokine that enhances skeletal muscle fatty acid oxidation via AMPK, and irisin, which promotes browning of white adipose tissue—both effects improving systemic insulin sensitivity [145]. Additionally, curcumin downregulates resistin and TNF-α expression in adipose tissue by interfering with NF-κB p65 nuclear translocation, thereby reducing local inflammation that impairs insulin action [143,145].

### 7.5. Conclusions and Future Directions

Curcumin, the principal bioactive curcuminoid of turmeric, has demonstrated robust glycemic control benefits across in vitro, animal, and human studies, acting through AMPK activation, enhancement of insulin signaling, suppression of hepatic gluconeogenesis, and modulation of gut microbiota. Clinical trials report that curcumin supplementation prevents progression from prediabetes to T2D, lowers fasting glucose, HbA1c, and HOMA-IR, and reduces inflammatory markers [125,126]. Also, meta-analyses confirmed significant improvement in metabolic parameters, especially due to glycemia control in response to curcumin intake [127,130]. However, the clinical utility of curcumin is uncertain due to its low oral bioavailability, variability in extract formulations, and the lack of long-term, large-scale human trials; thus, further studies in this area are warranted [115,136].

## 8. Conclusions

This review of the available literature indicates that many natural bioactive compounds show promising potential in improving glycemic control, including lowering blood glucose levels and enhancing tissue sensitivity to insulin. In vitro, in vivo, and clinical studies confirm the effectiveness of compounds such as MH, β-carotene, resveratrol, steviol glycosides, and curcumin in modulating key metabolic pathways, reducing oxidative stress—which helps neutralize the adverse effects of hyperglycemia and inflammation—and supporting pancreatic β-cell function. However, it should be emphasized that the effects of these compounds often depend on dosage, duration of supplementation, and individual patient characteristics (such as age, BMI, and comorbidities). In many cases, study results are inconclusive or limited to animal models, which significantly complicates extrapolation to human populations. Therefore, further well-designed randomized clinical trials are essential to clearly determine effective and safe therapeutic doses and to elucidate the molecular mechanisms of action of individual compounds. Only then can natural bioactive substances be realized as supportive agents in the treatment of carbohydrate metabolism disorders.

## Figures and Tables

**Figure 1 nutrients-18-00052-f001:**
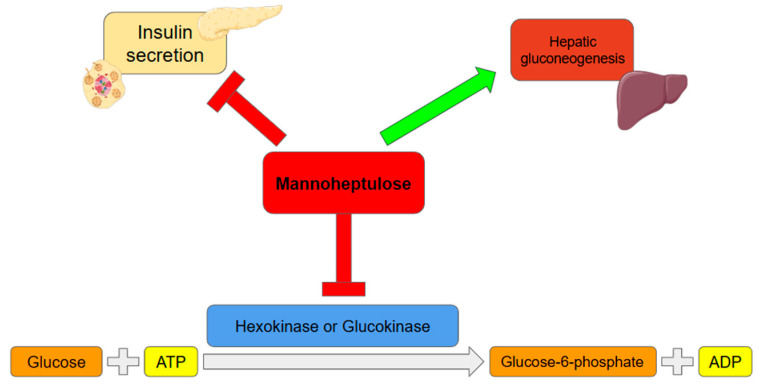
The role of mannoheptulose in glucose metabolism. Abbreviations: ADP—adenosine diphosphate; ATP—adenosine triphosphate. This figure was made using the Servier Medical Art collection (http://smart.servier.com/). Based on [9,14,30,31].

**Figure 2 nutrients-18-00052-f002:**
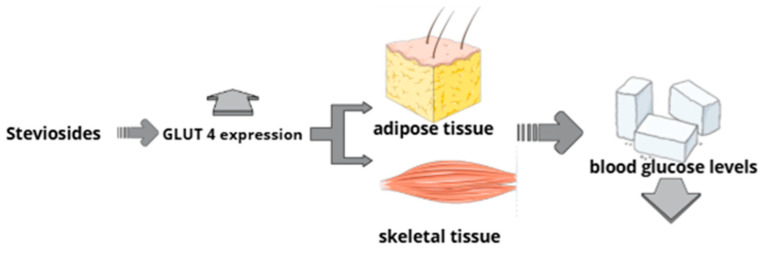
The proposed hypoglycemic mechanism of steviosides. Abbreviations: GLUT4—glucose transporter type 4. This figure was made using the Servier Medical Art collection (http://smart.servier.com/) and (https://www.canva.com/). Based on: [97].

**Table 1 nutrients-18-00052-t001:** Summary of results obtained from three research trials conducted on humans.

MH Form	MH Dosage	Effect on Glucose	Effect on Insulin	References
Fresh avocado fruit	2.15–12.83 g (33–200 mg/kg)	No significant changes	Significant decrease in insulin levels in 5 of 8 participants (*p* < 0.05)	[16]
MH solution (pilot + cross-over)	5–20 g/day	Increase of ~15% (1.5–4 h), return to normal after approx. 6 h	No increase, although sometimes a weakened early insulin response	[15]
AvX	10 g/day (~190 mg MH, ~2 mg/kg)	No significant changes in glucose AUC	Tendency to reduce insulin AUC, especially in participants with hyperinsulinemia	[12]

Abbreviations: AvX—avocado extract; AUC—area under the curve; MH—mannoheptulose.

**Table 2 nutrients-18-00052-t002:** Summary of β-carotene effect on glucose metabolism—in vitro studies.

Model	Dosage/Exposure	Effect on Glucose	Effect on Insulin	References
HepG2 insulin-resistant liver cells	15 μM β-carotene	↑ Glucose consumption (~1.49×), compared to metformin (0.1 μg/mL) (1.72×)	Insulin-like	[38]
L6 skeletal muscle cells	Not specified	↑ Glucose uptake (~1.4×), comparable to insulin (1.47×); low cytotoxicity	Insulin-mimetic	
Human adipocytes (co-treatment)	β-carotene + naringenin	↑ GLUT4, UCP1, adiponectin expression; ↑ PPARα, PPARγ, PGC-1α	↑ Insulin sensitivity, ↑ thermogenesis	[39]
Gestational diabetes in vitro model	β-carotene	↑ SHBG expression → ↑ GLUT4 → improved glucose transport	Improved insulin responsiveness	[40]

Abbreviations: ↑—increase; ↓—decrease; GLUT4—Glucose transporter type 4; HepG2—Human hepatocellular carcinoma cell line, clone 2; PPAR α and γ—Peroxisome Proliferator-activated receptor α and γ; PGC-1α—Peroxisome proliferator-activated receptor gamma coactivator 1-alpha; SHBG—Sex hormone-binding globulin; UCP1—Uncoupling protein 1.

**Table 3 nutrients-18-00052-t003:** Summary of β-carotene effects on glucose metabolism—animal trials.

Model	Dosage/Treatment	Effect on Glucose	Effect on Insulin	References
HFD-induced obese mice	β-carotene 3 mg/kg/day	↓ Hepatic fat, ↑ glucose utilization	↑ Insulin sensitivity	[42]
HFD mice + metformin	β-carotene + metformin	Synergistic effect: ↑ glucose oxidation	↑ Insulin signaling, ↑ metabolic gene expression	
Aged diabetic rats	β-carotene + Mg + Zn + metformin	Better glycemic control vs. metformin alone	↑ Insulin sensitivity	[43]
T2D mice fed Gac fruit aril	High in β-carotene	↓ FBG, ↑ glucose tolerance	↑ GLP-1 (~2×), ↑ β-cell function	[44]
GLP-1 receptor KO mice + Gac aril	High in β-carotene	Glycemia reduction	Confirms GLP-1-mediated mechanism	

Abbreviations: ↑—increase; ↓—decrease; FBG—Fasting blood glucose; GLP-1—Glucagon-like peptide-1; HFD—High-fat diet; KO—Knockout; T2D—Type 2 diabetes.

**Table 4 nutrients-18-00052-t004:** Summary of resveratrol effects on glucose metabolism—human studies.

Model	Dosage/Treatment	Effect on Glucose	Effect on Insulin	References
45 adults (30–70 y/o) with T2D, on hypoglycemics, BMI < 35	800 mg/day (2 × 400 mg), 8 weeks	Significant decrease in fasting glucose (−31.84 ± 47.6 mg/dL); no change in HbA1c	Not reported	[53]
24 obese males (18–70 y/o), BMI > 30	1500 mg/day (3 × 500 mg), 4 weeks	No significant change in glucose or HbA1c	Not reported	[68]
472 elderly patients (>60 y/o) with T2D	500 mg/day, 6 months	HbA1c decreased > 2%; improved SIRT1 and AMPK expression; increased G6Pase activity	Improved insulin sensitivity (based on SIRT1/AMPK pathway)	[56]
Meta-analysis: 921 subjects (45–59 y/o and ≥60 y/o)	250–500 mg/day as optimal dose	Significant improvement in glycemic control (↓ glucose, HbA1c)	Improved insulin sensitivity (↓ HOMA-IR), especially in ≥60 y.o. group	[69]
Meta-analysis: 871 T2D patients	≥500 mg/day	Significant improvement in glycemic control (↓ glucose, HbA1c)	Not reported	[70]

Abbreviations: ↓—decrease; AMPK—AMP-activated protein kinase; BMI—Body Mass Index; G6Pase—Glucose-6-phosphatase; HbA1c—Glycated hemoglobin; HOMA-IR—Homeostatic Model Assessment for Insulin Resistance; SIRT1—Sirtuin 1; T2D—type 2 diabetes.

**Table 5 nutrients-18-00052-t005:** A summary of the results obtained from human studies investigating the effects of stevia and steviosides on glycemia control.

Model	Dosage/ Treatment	Effect on Glucose	Effect on Insulin	References
20 adults	1 g/day powdered stevia leaf, 60 days	Significant decrease in FBG (form 156.61 ± 31.32 to 123.55 ± 22.94 mg/dL) and postprandial glucose (from 225.17 ± 43.86 to 200.60 ± 43.80 mg/dL); no change in control group	Not reported	[101]
150 participants: 40 with T2D, 60 obese, 50 healthy controls	4 mg/kg b.w./day, 24 weeks	Improved glycemic control, ↓ HbA1c, ↓ LDL, ↓ triglycerides; weight gain in obese group	Decreased fasting serum insulin; improved insulin sensitivity	[88]
Meta-analyses of 7 RCTs, including 462 participants with hypertension, diabetes, or hyperlipidemia	Various doses (200–1500 mg/day) and durations across studies (4 h–2 years)	No statistically significant changes in FBG or HbA1c, though a downward trend was observed	No reported	[102]
Meta-analyses of 26 RCTs including 1439 individuals with T2D, IR, hypertension, hyperlipidemia, or healthy subjects	Various doses (13.2–4000 mg/day) and durations across studies (1 day–2 years)	Significant reduction in FBG (WMD = −3.84; 95% CI: −7.15 to −0.53; *p* = 0.02), especially in individuals with high BMI or T2D	Some improvement in insulin sensitivity in subjects with T2D or high BMI	[103]

Abbreviations: ↓—decrease; BMI—body mass index; CI—confidence interval; FBG—fasting blood glucose; HbA1c—glycated hemoglobin; LDL—low-density lipoprotein; RCT—randomized controlled trial; T2D—type 2 diabetes; WMD—weighted mean difference.

**Table 6 nutrients-18-00052-t006:** A summary of clinical studies evaluating the effects of curcumin on glucose metabolism in humans.

Model	Dosage/Treatment	Effect on Glucose	Effect on Insulin	References
240 prediabetic subjects, 9-month double-blind RCT	Curcumin extract, 750 mg twice daily (9 months)	Prevented T2D development (0% vs. 16.4% in placebo, *p* < 0.001); improved glucose regulation across 3, 6, and 9 months	↑ HOMA-β (61.58 vs. 48.72; *p* < 0.01); ↓ HOMA-IR (3.22 vs. 4.04; *p* < 0.001); ↑ adiponectin (22.46 vs. 18.45; *p* < 0.01)	[125]
100 T2D patients; EPA + nano-curcumin co-administration	EPA (500 mg) + nano-curcumin (80 mg) daily, 12 weeks	Reduced hs-CRP; improved TAC; overall metabolic improvement	↓ Insulin [MD: −1.44 (−2.70, −0.17)]	[126]
Meta-analysis of 59 RCTs	Curcumin/turmeric ≥ 500 mg/day; ≥12 weeks	↓ Fasting glucose (WMD −4.60 mg/dL; 95% CI −5.55, −3.66); ↓ HbA1c (WMD −0.32%; 95% CI −0.45, −0.19)	↓ Fasting insulin (WMD −0.87 µIU/mL; 95% CI –1.46, −0.27); ↓ HOMA-IR (WMD −0.33; 95% CI −0.43, −0.22)	[127]
Women with PCOS	500–1500 mg/day, 6–12 weeks	↓ Fasting glucose (MD: −2.77; 95% CI: −4.16, −1.38); improved lipid profile	↓ Fasting insulin (MD: −1.33; 95% CI: −2.18, −0.49); ↓ HOMA-IR (MD: −0.32; 95% CI: −0.52, −0.12)	[128,129]
Umbrella review of 22 meta-analyses	Various curcumin doses	↓ FBG and HbA1c	↓ Insulin and HOMA-IR	[130]

Abbreviations: ↑—increase; ↓—decrease; EPA—eicosapentaenoic acid; FBG—fasting blood glucose; HbA1c—glycated hemoglobin; HOMA-β—homeostasis model assessment β-cell function; HOMA-IR—homeostasis model assessment of insulin resistance; hs-CRP—high-sensitivity C-reactive protein; PCOS—polycystic ovary syndrome; TAC—total antioxidant capacity; T2D—type 2 diabetes.

## Data Availability

Not applicable.

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
