# Peer review of "Influence of Certain Natural Bioactive Compounds on Glycemic Control: A Narrative Review"

_nutrients, 2025, doi:10.3390/nu18010052_

Round 1

Reviewer 1 Report

Comments and Suggestions for Authors

Dear Authors,

After reviewing the following manuscript entitled ‘Influence of natural bioactive compounds on glycemic control - a narrative review’ (nutrients-3989577), I affirm that it is an interesting and useful work for those who wish to approach knowledge about certain natural bioactive compounds involved in blood glucose regulation.

The abstract is well written and introduces the topic to the reader by summarising the objectives of the review and the limitations of the research regarding the compounds treated for managing glycemic disorders.

However, after careful consideration, I would like to suggest that the manuscript undergo minor revisions before it can be reconsidered for publication. Please find my detailed comments and recommendations in the attached file

Kind regards

Author Response

Dear Reviewer,

Please find attached the author's response to review.  

Kind regards

Reviewer 2 Report

Comments and Suggestions for Authors

The review provides a comprehensive summary on the glycemic control properties of some bioactive compounds. The ms is suitable for publication ondition that some points are addressed by the Authors.

  1. Mannoheptulose

The section titled "Unripe Avocado Extract – Mannoheptulose" primarily discusses Mannoheptulose. However, the comparative analysis with conventional medications draws extensively on results from leaf and seed extracts of Persea americana. The anti-diabetic activity of these extracts is often attributed to compounds other than MH, such as polyphenols and flavonoids. The authors should more explicitly clarify that the pharmacological equivalence found in comparison studies (e.g., leaf extracts vs. glibenclamide) relies on the pleiotropic effects of the whole P. americana extracts, while distinguishing this from the specific mechanism of MH. Moreover, the review should clearly distinguish the long-term efficacy of MH based on species: in Par. 3.5, the statement regarding the association between higher doses of MH and reduced blood glucose levels should be qualified as being based solely on animal models.

  1. β-Carotene

Given that β-Carotene’s hypothesized anti-diabetic mechanisms are only at the pre-clinical level, line 478 should change to: "In preclinical studies (in vitro and animal trials), β-carotene supplementation has been shown to improve FBG, insulin sensitivity, and β-cell function."

  1. Resveratrol

The manuscript cites the randomized controlled trials (RCTs) by Khodabandehloo et al. and Poulsen et al. However, these individual studies are then incorporated into the conclusions derived from the after mentioned in the text meta-analyses by García-Martínez. When discussing the overall findings of the meta-analyses, the authors should explicitly write that the systematic review integrates the data from trials individually reported before in the text.

  1. Steviosides

The review references human studies on steviosides, including the study by Mathur Ritu and Johri Nandini (2016) and the controlled clinical trial by Rashad et al. (2019). The former study is an experimental case study conducted on 20 subjects with type 2 diabetes, divided into two groups (10 control and 10 experimental). There is no mention of registration in clinical trial databases (e.g., ClinicalTrials.gov), nor is it indicated as randomized or double-blind. It is closer to an observational pilot study with an intervention. Authors should report this. With respect to the RCT by Rashad et al. (2019), which investigated stevioside supplementation in T2D and obese patients, this study reported improved glycemic control. However, the methodology employed in Rashad presents a High Risk of Bias, making the reported positive effects unreliable for supporting efficacy claims. In particular, the study design compared an Intervention group (100 subjects: 40 T2D patients, 60 obese patients) with a Control group (50 healthy subjects). This non-random allocation and the comparison between a sick group receiving treatment and a healthy group receiving no equivalent intervention fundamentally undermines the ability to attribute observed changes solely to the stevioside. Moreover, over the 24-week intervention period, the Intervention group showed significantly higher values in total caloric intake, total carbohydrates, total protein, and total fat after stevioside supplementation compared to their baseline diet. Since total energy intake and macronutrient consumption are primary determinants of glycemic control and weight, it is impossible to confidently distinguish the physiological effect of the stevioside from the profound confounding effect of this substantial increase in caloric and macronutrient intake. Authors should explicitly acknowledge these severe methodological flaws, and the reported improvements in glycemic control from Rashad et al. should be treated as low-certainty evidence or results that are highly susceptible to confounding due to inadequate study design and significant differences in nutrient intake between the groups, rather than reflecting the pure pharmacological effect of the stevioside.

Author Response

(The authors gave the same response as above.)

Reviewer 3 Report

Comments and Suggestions for Authors

This is a narrative review of active compounds with potential blood glucose regulating activity, in which the available and valuable information on in vitro, in vivo, and clinical studies on compounds such as mannoheptulose, β-carotene, resveratrol, steviol glycosides, and curcumin is adequately described.

However, in my opinion there are some aspects in the abstract, introduction and conclusions that should be clarified, and where appropriate, improved.

Comments, questions and/or suggestions

1.- 13-14  The authors highlight the existence of effective therapeutic strategies for the control of glycemic disorders, which, however, are associated with limited patient tolerance and adherence.

It is suggested to briefly add the terms in which the authors consider the effectiveness of glycemic control disorders; i.e. glycemic control according to goals, reduction in the incidence of complications, mortality, etc.

2.- 37-38 The authors point out that as the prevalence of glycemic disorders increases, there is also a rising number of deaths associated with T2D.

This would seem to indicate that the effectiveness of therapeutic strategies for controlling glycemic disorders has an intrinsic limit, which is 15 often associated with limited patient adherence or tolerance. It is suggested that the appropriateness of making a change in this regard be evaluated.

3.- 16-17 The authors point out there is growing interest in natural bioactive compounds that may support glycemic regulation while posing a lower risk of adverse effects compared to synthetic drugs.

This assertion would seem to indicate that natural bioactive compounds would, by that fact, possess a lower risk of adverse effects compared to synthetic drugs, which clearly sounds pharmacologically incorrect.

4.- 50-53 The authors highlight the concept of caloric restriction mimetics (CRM), as substances that replicate the metabolic, hormonal, and physiological effects associated with reduced calorie intake without requiring significant decreases in food consumption.

Clarify whether mannoheptulose, β-carotene, resveratrol, steviol glycosides, and curcumin are considered and postulated by the authors as CRM or as natural substances with anti-hyperglycemic activity?.

 5.- 47-49 The authors highlight that caloric restriction (CR) is considered one of the effective strategies for preventing and treating excessive body weight, often correlated with carbohydrate dysregulation. However, adherence to strict dietary guidelines often proves challenging for patients with established poor dietary habits.

Would the authors consider CRMs to be particularly useful in patients with insulin resistance, prediabetes, obesity, or diabetes who have difficulty adhering to prescribed lifestyle changes such as diet and exercise? Please clarify.

Lifestyle modifications based on diet and exercise are considered the cornerstone of weight and blood glucose regulation. Would CRMs aim to achieve this without requiring significant lifestyle changes?

It is suggested that the appropriateness of making a changes in this regard be evaluated.

Author Response

(The authors gave the same response as above.)

Round 2

Reviewer 3 Report

Comments and Suggestions for Authors

No additional comments